

# Invariants of winding-numbers and steric obstruction in dynamics of flux lines

**Olivier Cépas[1*] and Peter M. Akhmetiev[2,3]**

**1** Université Grenoble Alpes, CNRS, Grenoble INP, Institut Néel, 38000 Grenoble, France
**2** HSE Tikhonov Moscow Institute of Electronics and Mathematics,
34 Tallinskaya Str., 123458, Moscow, Russia
**3** Pushkov Institute of Terrestrial Magnetism, Ionosphere and
Radio Wave Propagation, Kaluzhskoe Hwy 4, 108840, Moscow, Troitsk, Russia

⋆ olivier.cepas@neel.cnrs.fr

## Abstract

We classify the sectors of configurations that result from the dynamics of 2d crossing flux lines, which are the simplest degrees of freedom of the 3-coloring lattice model. We show that the dynamical obstruction is the consequence of two effects: (i) conservation laws described by a set of invariants that are polynomials of the winding numbers of the loop configuration, (ii) steric obstruction that prevents paths between configurations, for lack of free space. We argue that the invariants fully classify the configurations in five, chiral and achiral, sectors and no further obstruction in the limit of low-winding numbers.



# 1   Introduction

Quantities conserved by the dynamics, or more generally, invariants of transformations, are essential to classify configurations and states of matter. They provide us with shortcuts to identify configurations up to "unessential" deformations, by giving them a common label. Examples of invariants range from basic symmetry operations (*e.g.* vector lengths for rotations), to complex problems (knots, homotopy groups...), or appear unexpectedly in some popular games (14-15 puzzle, Rubik's cube, Solitaire...) [1]. In the physical context, we generally label the physical states according to the irreducible representations of a symmetry group. However, the internal symmetries may be unknown or unexpected (*e.g.* SO(4) for the hydrogen atom). The particle labels of a field theory at a given scale may thus be the result of non-obvious invariants of the dynamics of more "elementary" degrees of freedom at a lower scale. Finding invariants in the dynamics of simple degrees of freedom is therefore of particular interest, as it reveals some form of internal symmetry.

Here we study the dynamics of a locally-constrained classical model, and provide the invariants that classify its sectors. Locally-constrained models (also called "vertex" or "ice"-type) can be viewed as the strong-coupling limit of physical models, in that the strong local interactions are replaced by hard on-site constraints. Among them, there are examples where natural dynamical transformations lack ergodicity [2–9]. This is the case of the 3-coloring model, which we consider here. In this model, a hard local constraint prevents neighboring edges to have the same color among three, on a regular hexagonal lattice (see Fig. 1 for an example). It was originally introduced as an exactly-solvable model of ice-type where the entropy could be explicitly calculated [10]. The model was argued later to describe some magnetic materials, having the geometry of the kagome lattice[1], where the three colors are three spin directions that are forced to be different by a strong local antiferromagnetic interaction [2, 11, 12]. As a simple model, it may have various other physical applications, from a minimal description of glasses [13–15], to superconducting arrays [14]. The degrees of freedom that are compatible with the constraints are loops of alternating colors (see Fig. 2), similar to the moves Kempe introduced to swap colors in planar maps. When periodic boundary conditions are used, this dynamics is known to be nonergodic, for reasons not fully understood [2–5,9]. The color configurations can be put in equivalence classes, called Kempe sectors, if they are connected by

---

[1]The sites of the kagome lattice are the centers of the edges of the hexagonal lattice.

the dynamics, the number of which, $n_K > 1$ [4,5], has been enumerated numerically on small random cubic graphs [16], or regular hexagonal clusters [4,17]. An invariant has been found, allowing to distinguish odd from even colorings [4,5,17]. It does not exhaust the number of sectors [17], so that the classification of Kempe sectors is so far incomplete.

With periodic boundary conditions, closed loops are characterized by winding numbers which count the number of times the loops wind around the boundaries[2]. The dynamics is twofold. First, small local loops locally deform the winding loops, but preserve their topological winding numbers and thus define homotopy classes (flux conservation). Second, the dynamics of the winding loops do not conserve the topological numbers (flux insertion). Since they are integer numbers, this results in a dynamics described by transformations from integers to integers. The question arises as to whether this integer dynamics can reach any configuration of winding numbers. In fact, it cannot and there are invariants, stable with system size, that classify the Kempe sectors in terms of polynomials of the winding numbers (which are not individually conserved). While many sectors remain undistinguished by these invariants, we will further argue that there is no other invariant, *i.e.* this set is complete. In fact, paths between these additional sectors involve intermediate configurations that exist at larger sizes: these sectors at a given size are isolated because of steric obstruction.

In addition to the stable classification of the configurations, these results show the existence of many disconnected sectors of flux lines at fixed size. In the context of the "slow" dynamics of glasses, nonergodicity plays a central role. It serves as a basic property to generate new timescales, since the reconnection of these sectors may be assured by slower dynamical processes. This is what happens here as a consequence of the steric obstruction. On the technical side, these questions are relevant for Monte Carlo simulations where nonlocal moves are used to sample the space of configurations [18]. Invariants introduce a bias that has to be overcome, either by giving up the periodic boundary conditions [3], or by enriching the moves with that of "stranded" loops [17].

The paper is organized as follows. In section 2, we recall the direct numerical construction of Kempe sectors on small clusters by constructing the 3-colorings and the dynamics between them exhaustively. In section 3, we define the winding loop configurations and the effective dynamics that transform the winding numbers. By iterating this dynamics, we construct the Kempe sectors of winding numbers for much larger systems. We then construct three invariants of the dynamics in section 4, stable at all system sizes. We argue in section 5 that this set of invariants is complete: the remaining sectors result from steric constraints and are absent when such constraints are relaxed (section 5).

## 2 Kempe sectors by direct construction

The model consists of coloring the edges of a regular hexagonal lattice of linear size $L$ (and $N = 3L^2$ edges) with three colors, *e.g.* A, B, C, such that each vertex has three edges colored with three different colors, see Fig. 1. The number of such 3-colorings has been calculated exactly by Baxter [10] and scales as $\sim 1.1347^N$ in the thermodynamic limit. Periodic boundary conditions are used, so the graph has the geometry of a torus and homotopy classes exist.

The successive edges of two colors, say B and C (or A and B, or A and C) form self-avoiding closed loops. All loops are fully-packed. In Fig. 2, all B-C loops are shown and edges colored A have been withdrawn. Exchanging the two colors of one of these loops gives a new valid 3-coloring (all edges colored with three different colors). It is an example of Kempe move. There are different types of loops, $a$-type (B-C), $b$-type (A-C) and $c$-type (A-B). When a loop of one type flips, it reorganizes the other types of loops. The loops are also characterized by the

---

[2]Winding numbers label the homotopy classes of the torus. They measure the 2d flux through its two cycles.

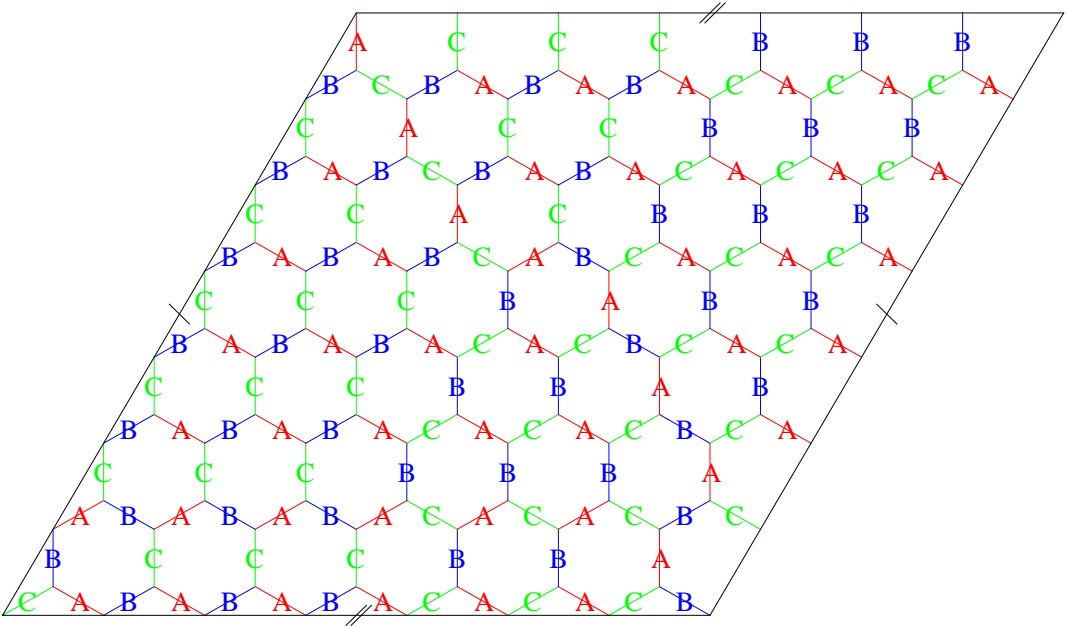

Figure 1: Example of a 3-coloring of the $N = 3L^2$ edges of a hexagonal lattice of linear size $L = 7$. Periodic boundary conditions are used to form a torus and define homotopy classes.

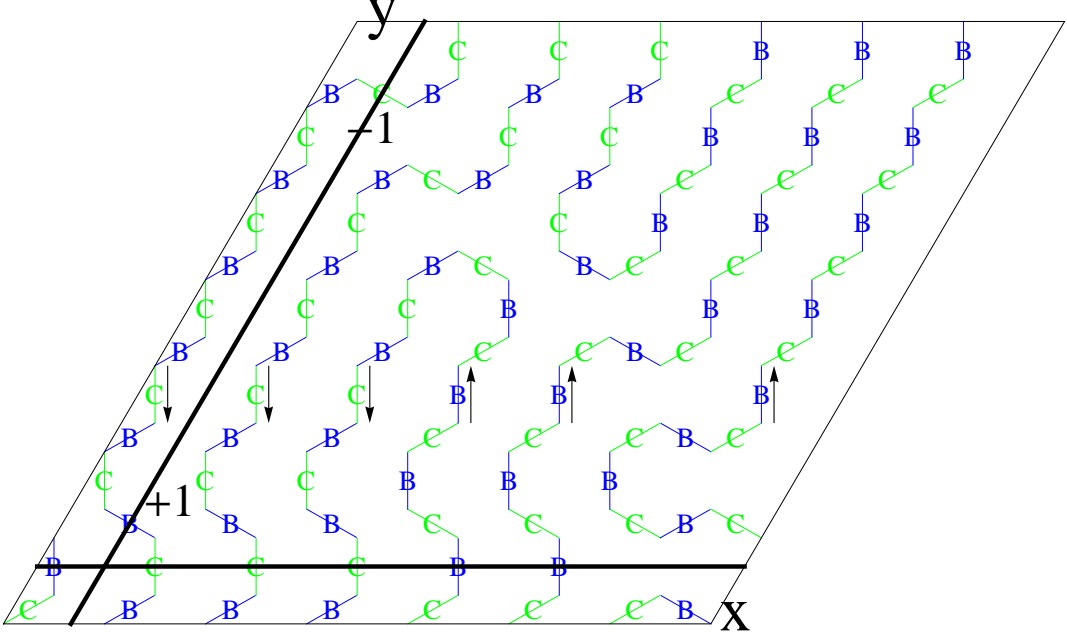

Figure 2: Same configuration as in Fig. 1 with red edges not shown, emphasizing closed self-avoiding B-C (blue-green) loops, which can be seen as flux lines through the $x$ and $y$ lines. Orientation of the loops is defined in Fig. 3. There are three types of flux lines (not shown) which cross.

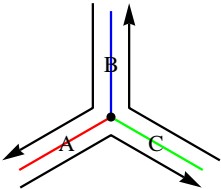

Figure 3: Definition of the orientation of the loops for a black vertex (the definition is opposite on white vertices).

number of times they wind across the two boundaries, the winding numbers $(p,q)$ (homotopy classes), the precise definition of which will be given in 3.1.

For completeness, we recall the construction of Ref. [17]. We first construct numerically all 3-colorings of a cluster of size $L$. This procedure is obviously limited by the exponentially-growing number of states to small $L \lesssim 8$. By iteration of the Kempe moves, considering the motion of *all* loops ($a$, $b$, $c$ type, winding and nonwinding), we find that not all states can be connected but fall into closed disconnected sectors, called Kempe sectors. The number of Kempe sectors, $n_K$, depends on $L$ and is given in table 1. It is increasing with system size, but it is difficult to know whether it is infinite in the thermodynamic limit. We also give the number of 3-colorings in each Kempe sector $Z_{i=1,\dots,n_K}$, their multiplicity, and the labels obtained in section 4.

Table 1: Number of Kempe sectors $n_K$ and number of 3-colorings in each sector $Z_{i=1,\dots,n_K}$ (sector multiplicity is indicated). The labels given in brackets are the purpose of the paper (defined in section 4). For $L = 7$ and $L = 8$, they do not exhaust the number of sectors, see section 5.

| $L$ | $n_K$ | $Z_{i=1,\cdots,n_K}, I$ |
|---|---|---|
| 2 | 1 | 24 (0) |
| 3 | 2 | 60 (0), 60 (1,-1,1) |
| 4 | 2 | 240 (0), 1920 (1,-1,-1) |
| 5 | 3 | 35340 (0), 276 (1,-1,-1), 13800 (1,-1,1) |
| 6 | 4 | 2879856 (0), 307296 (1,-1,1), 19440(×2) (1,1,±1) |
| 7 | 2761 | 50683920 (0), 1140 (1,-1,1), 424598328 (1,-1,-1), 6 (×2758) (1,1,±1) |
| 8 | 9 | 59539228896 (0), 54178583040 (1,-1,1), 14555136 (1,-1,-1), 28369152(×6) (1,1,±1) |

The dynamics of nonlocal winding loops do not conserve the winding numbers, so that homotopy classes are not preserved. Nevertheless, they appear to be sets of winding numbers that remain disconnected from other sets of winding numbers. Importantly, disconnected Kempe sectors having the same winding numbers do not occur, in general. There is one exception in the range of sizes available. For $L = 7$, a large number of Kempe sectors is found (table 1), all having the minimum number of 6 states. These are special 3-colorings, characterized by a single loop of each type, of maximal size, $2N/3$ [17]. A translation of these loops does not change the winding numbers but cannot be connected, thus generating many special sectors. Except for this size, an homotopy class is found to be connected.

We study the main obstruction, stable with system size, the one related to the winding numbers (keeping in mind that there may be finer obstructions for special $L$), by setting up a dynamics for the winding numbers themselves.

# 3 Integer dynamics of winding numbers

The aim of this section is to set up the dynamics of the winding numbers, when nonlocal moves are allowed, *i.e.* the set of transformations from integers to integers. We next study its ergodic properties, and what classes form.

## 3.1 Definition of winding numbers

The total winding numbers of a configuration are defined by a triplet of vectors $(\boldsymbol{a}, \boldsymbol{b}, \boldsymbol{c})$,

$$\boldsymbol{a} = \sum_l (a_l^x, a_l^y), \tag{1}$$

$$\boldsymbol{b} = \sum_l (b_l^x, b_l^y), \tag{2}$$

$$\boldsymbol{c} = \sum_l (c_l^x, c_l^y), \tag{3}$$

where $\boldsymbol{a}$, $\boldsymbol{b}$, $\boldsymbol{c}$ are two-component vectors (we will define a third component below) corresponding to the three types of loops, $a \equiv$ B-C, $b \equiv$ C-A and $c \equiv$ A-B loops. The sum is over all loops $l$. For instance, $a_l^\alpha$ is the winding number of the $l$ loop across a cycle $\alpha = x$ or $y$ of the torus (Fig. 2). They count the number of geometrical (signed) crossings. To compute them, we first orient the loops. The hexagonal lattice is bipartite and we can define two sets of vertices, black and white, connected by the edges. We then orient the loops from B to A, A to C and C to B on black vertices (the opposite for white vertices), as shown in Fig. 3. With this convention of orientation, every B-colored edge that cuts the $y$-axis (and is part of a B-C loop), contributes to the winding number by $+1$ and every edge colored C contributes by $-1$. In the example of Fig. 2, $a_l^y = 1 - 1 = 0$ for the loop $l$ that crosses the $y$-axis in two places and $a_l^y = 0$ for the other loops which do not cross the $y$-axis, so that $a_y = 0$ (there is no flux through the $y$-axis). Therefore, if we denote by $N_B^y$ and $N_C^y$ the number of edges along the $y$ line that are colored B and C, we get $a_y = N_B^y - N_C^y$. Similarly, we define a set of numbers $N_i^\alpha$, $i = A, B, C$ and $\alpha = x, y$, so that

$$a_x = N_B^x - N_C^x, \tag{4}$$
$$a_y = N_B^y - N_C^y, \tag{5}$$
$$b_x = N_C^x - N_A^x, \tag{6}$$
$$b_y = N_C^y - N_A^y, \tag{7}$$
$$c_x = N_A^x - N_B^x, \tag{8}$$
$$c_y = N_A^y - N_B^y. \tag{9}$$

Since we have the constraints on the number of colors, $0 \leq N_i^\alpha \leq L$, we get some constraints for the winding numbers,

$$|a_\alpha| \leq L, \qquad |b_\alpha| \leq L, \qquad |c_\alpha| \leq L, \tag{10}$$

where $\alpha = x, y$. It is also apparent that,

$$\boldsymbol{a} + \boldsymbol{b} + \boldsymbol{c} = 0. \tag{11}$$

A consequence is that it is not possible to have a single winding loop. Indeed, in this case, we would have (without loss of generality) $\boldsymbol{a} = (p, q) \neq 0$ and $\boldsymbol{b} = \boldsymbol{c} = 0$ and the sum would not be zero. Note that a permutation of all colors B and C is an exchange of the winding numbers $\boldsymbol{b}$ and $\boldsymbol{c}$ with a global sign change, $\boldsymbol{a} \to -\boldsymbol{a}$ and $\boldsymbol{b} \to -\boldsymbol{c}$, $\boldsymbol{c} \to -\boldsymbol{b}$.

Since all of the $L$ edges cutting the lines $\alpha = x, y$ have some color, we have,

$$\sum_i N_i^\alpha = L, \tag{12}$$

for all $\alpha$. It is convenient to define another set of numbers $N_i^z$, $i = A, B, C$, along a third line in the $z$ direction (the $y, z$ axis are at $\pm 120^o$ with the $x$ axis). Indeed, the sum of the three numbers of a given color times $L$ is the total number of this color on the graph, *e.g.* $L(N_A^x + N_A^y + N_A^z) = N/3 = L^2$, and similarly for B and C:

$$\sum_\alpha N_i^\alpha = L, \tag{13}$$

for all $i = A, B, C$ ($N_A^z$ is not an independent number). Since $N_A^z$ must be positive and smaller than $L$ it gives three additional constraints: $N_i^x + N_i^y \le L$.

Given the two equations (12) and (13), we can invert Eqs. (4)-(9). It is convenient to define a third component $a_z$ such that $a_x + a_y + a_z = 0$ (and similar definitions for $b$ and $c$). We can write the relations between $(N_A, N_B, N_C)$ and $(a, b, c)$ in the form,

$$N_A = \frac{L}{3}\mathbf{1} + \frac{1}{3}(c - b), \tag{14}$$

$$N_B = \frac{L}{3}\mathbf{1} + \frac{1}{3}(a - c), \tag{15}$$

$$N_C = \frac{L}{3}\mathbf{1} + \frac{1}{3}(b - a), \tag{16}$$

where $\mathbf{1} = (1, 1, 1)$ and all vectors have now three (non-independent) components. There is a central sector, called the "0-sector" characterized by the absence of winding loops, $(a, b, c) = 0$, which exists when $L$ is a multiple of three. A finite $(a, b, c)$ can be seen as a deviation from this sector, but the integers must be chosen such that the $N_\alpha$ themselves are integers, *i.e.* $b - c$, $c - a$, and $a - b \equiv L (\mathrm{mod}\ 3)$. To have the same set of integers $(a, b, c)$ at $L$ and $L_1$, we must have $L_1 \equiv L (\mathrm{mod}\ 3)$. In particular, changing the sign of $(a, b, c) \to -(a, b, c)$ is possible only when $L$ is a multiple of three, since $b - c$ or $c - b$ are both multiple of three, in this case.

We can define a norm for the winding numbers,

$$n^2 = \frac{1}{6}(|a|^2 + |b|^2 + |c|^2), \tag{17}$$

where $|a|^2 = a_x^2 + a_y^2 + a_z^2$. It has a minimum at $n = 0$ in the "0-sector" and a maximum at $n = L$. The conditions $|a| \ll L$, $|b| \ll L$, and $|c| \ll L$ (or $n \ll L$) define the limit of low winding numbers. It can be seen as a *dilute* limit, since low-winding number loops may occupy a number of sites of order $L$, *i.e.* a *vanishing* fraction of the lattice sites. On the contrary, when some winding numbers are a fraction of $L$, the loops occupy a *finite* fraction of the lattice sites (*finite-density* configurations).

A color configuration is thus reduced to a set of nine winding numbers $(a, b, c)$ or, equivalently, $(N_A, N_B, N_C)$. Only four of them are independent (given Eq. (11) and the definition of the $z$-components). A color configuration should satisfy the "box" constraints,

$$0 \le N_i^\alpha \le L, \tag{18}$$

where $i = A, B, C$ and $\alpha = x, y, z$. Given these constraints, the actual number of winding number configurations is a fraction of $L^4$, and, of course, much smaller than the number of color configurations.

## 3.2 Setting up the transformations of winding numbers

The dynamics we consider consists of exchanging the two colors A-B, B-C or C-A along any closed loop $l$. This is the simplest dynamics that preserves the constraints.

We now construct a "coarse-grained" dynamics in the much smaller space of winding number configurations. A nonwinding loop modifies locally the shapes of the winding loops but does not change the set of numbers $(\boldsymbol{a}, \boldsymbol{b}, \boldsymbol{c})$.

Let us consider a winding loop of type $a$ (B-C) with $\boldsymbol{a}_l = (1, 0)$. The loop winds around the $x$-axis in the $y$ direction. It intersects the $x$-axis in a site with color B, given the convention of orientation. The flip of this loop changes the color on the $x$-axis from B to C. $a_x = N_B^x - N_C^x$ becomes $a_x - 2$. Simultaneously, $b_x = N_C^x - N_A^x = b_x + 1$ and $c_x = N_A^x - N_B^x = c_x + 1$. More generally, if the winding number is $a_x = w > 0$, there are $w$ sites along the $x$-axis which are in color B and which change to C after the flip, so that $a_x = N_B^x - N_C^x \to a_x + 2w = a_x + 2a_l^x$, and $b_x$ or $c_x$ change according to $b_x = N_C^x - N_A^x = b_x - w = b_x - a_l^x$. The same reasoning applies for the $y$ component, so that for a general flip of a loop with winding numbers $\boldsymbol{a}_l \to -\boldsymbol{a}_l$, we have the transformation law,

$$\boldsymbol{a}' = \boldsymbol{a} - 2\boldsymbol{a}_l, \tag{19}$$

$$\boldsymbol{b}' = \boldsymbol{b} + \boldsymbol{a}_l, \tag{20}$$

$$\boldsymbol{c}' = \boldsymbol{c} + \boldsymbol{a}_l. \tag{21}$$

$\boldsymbol{a}_l$ is parallel to $\boldsymbol{a}$, because all winding loops of a given type are self-avoiding (if $\boldsymbol{a}_l$ were not parallel to $\boldsymbol{a}$, we would have $\boldsymbol{a} \times \boldsymbol{a}_l \neq 0$ and they would cross). Let us define a *primitive* integer vector $\hat{\boldsymbol{a}} = (\hat{a}_x, \hat{a}_y)$ parallel to $\boldsymbol{a}$ but with the smallest integer coefficients,

$$\hat{\boldsymbol{a}} = \frac{\boldsymbol{a}}{\gcd(a_x, a_y)}, \tag{22}$$

where $\gcd(x, y) > 0$ is the greatest common factor of $x$ and $y$. The primitive vector $\hat{\boldsymbol{a}}$ characterizes the elementary winding loops, and $\gcd(a_x, a_y)$ is the number of such loops. We can choose $\boldsymbol{a}_l = k\hat{\boldsymbol{a}}$ with $k$ a positive or negative integer:

$$(\boldsymbol{a}', \boldsymbol{b}', \boldsymbol{c}') = (\boldsymbol{a} + 2k\hat{\boldsymbol{a}}, \boldsymbol{b} - k\hat{\boldsymbol{a}}, \boldsymbol{c} - k\hat{\boldsymbol{a}}). \tag{23}$$

In terms of the number of colors, we have $(N_A', N_B', N_C') = (N_A, N_B + k\hat{\boldsymbol{a}}, N_C - k\hat{\boldsymbol{a}})$, which makes explicit that a flip of a $a$-type loop exchanges the B and C colors along the different cycles. There are two other similar transformations on A-B or A-C loops.

In summary, the dynamics consists of three transformations applied on the set of winding numbers,

$$T_a(k) : (\boldsymbol{a}', \boldsymbol{b}', \boldsymbol{c}') = (\boldsymbol{a} + 2k\hat{\boldsymbol{a}}, \boldsymbol{b} - k\hat{\boldsymbol{a}}, \boldsymbol{c} - k\hat{\boldsymbol{a}}), \tag{24}$$

$$T_b(k) : (\boldsymbol{a}', \boldsymbol{b}', \boldsymbol{c}') = (\boldsymbol{a} - k\hat{\boldsymbol{b}}, \boldsymbol{b} + 2k\hat{\boldsymbol{b}}, \boldsymbol{c} - k\hat{\boldsymbol{b}}), \tag{25}$$

$$T_c(k) : (\boldsymbol{a}', \boldsymbol{b}', \boldsymbol{c}') = (\boldsymbol{a} - k\hat{\boldsymbol{c}}, \boldsymbol{b} - k\hat{\boldsymbol{c}}, \boldsymbol{c} + 2k\hat{\boldsymbol{c}}), \tag{26}$$

where $k$ is an arbitrary positive or negative integer. The transformations are such that $\boldsymbol{a}' + \boldsymbol{b}' + \boldsymbol{c}' = 0$. Since, among the nine integers $(\boldsymbol{a}, \boldsymbol{b}, \boldsymbol{c})$, only four are independent, the transformations are maps from $\mathbb{Z}^4$ onto $\mathbb{Z}^4$.

### 3.2.1 First example: permutation of colors

Consider a transformation $T_a(k_a)$ with $k_a = -\gcd(a_x, a_y)$, i.e. $k_a \hat{\boldsymbol{a}} = -\boldsymbol{a}$:

$$T_a(k_a) : (\boldsymbol{a}', \boldsymbol{b}', \boldsymbol{c}') = (-\boldsymbol{a}, -\boldsymbol{c}, -\boldsymbol{b}), \tag{27}$$

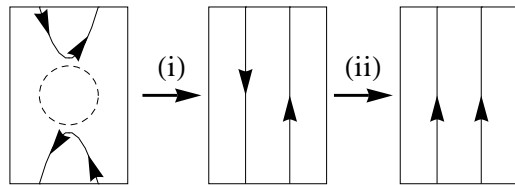

Figure 4: Creation of two opposite winding loops after the flip of the dashed loop (i) and insertion of flux after the flip of the winding loop (ii).

since $a + b + c = 0$. From Eqs. (14)-(16), we see that $(N'_A, N'_B, N'_C) = (N_A, N_C, N_B)$, which is a permutation of B and C. We have six such permutations, generated by $P_i \equiv T_i(k_i)$, $i = a, b, c$,

$$(a, b, c), -(a, c, b), -(c, b, a), -(b, a, c), (b, c, a), (c, a, b).$$

Note that the triplet $(a, b, c)$ acquires a sign according to whether the permutation is even or odd (it transforms like the signature representation).

### 3.2.2 Second example: insertion of flux

Suppose that $\gcd(a_x, a_y) = 1$ and consider,

$$T_a(1) : (a', b', c') = (3a, b - a, c - a), \tag{28}$$

which describes a new configuration with three parallel winding loops of type $a$, since $\gcd(a'_x, a'_y) = 3$. It becomes in terms of number of colors,

$$(N'_A, N'_B, N'_C) = (N_A, 2N_B - N_C, 2N_C - N_B).$$

Physically, it corresponds to the insertion of flux after creation of two opposite winding loops (Fig. 4). They must be parallel to the existing ones (*i.e.* noncrossing), which is taken care of, since the increase is along the direction of $a$. There must be room for them. The second condition leads to some constraints (see below).

### 3.2.3 Composition of transformations

The transformations do not commute in general. We have some useful relations,

$$T_i(-1).T_i(1) = \text{Id}, \tag{29}$$
$$T_i(1).T_i(1) = T_i(2), \tag{30}$$
$$T_i(1).T_i(-1) = T_i(-2), \tag{31}$$

where Id is the identity and $i = a, b, c$. This allows to restrict to $k = \pm 1$ and iterate.

### 3.2.4 Constraints

The transformations Eqs. (24)-(26) are not always possible because they may violate the space constraints:

$$0 \le (N_i^\alpha)' \le L, \tag{32}$$

where the prime denotes the numbers after the transformation. If one of these conditions is violated, the transformation is forbidden.

The transformations Eqs. (24)-(26) together with the constraints Eq. (32) constitute the "laws of dynamics" for the winding numbers. We have "coarse-grained" the dynamics by eliminating the effects of the dynamics of the local loops.

### 3.3 Numerical construction of Kempe sectors of winding numbers

We start by constructing all initial sets of integers $(a, b, c)$ for a given $L$ satisfying the basic constraints [Eqs. (14)-(16),(18)]. There are less than $L^4$ configurations, but more that the number of allowed winding-number classes in the coloring problem, so some of them are unphysical. While in the continuum limit we would expect all possible winding numbers, here the discreteness of the lattice gives additional constraints. Similarly, all winding numbers should be realized on the lattice but in the dilute limit[3]. Away from this limit, the set of configurations is enlarged, but we can still get some qualitative insights.

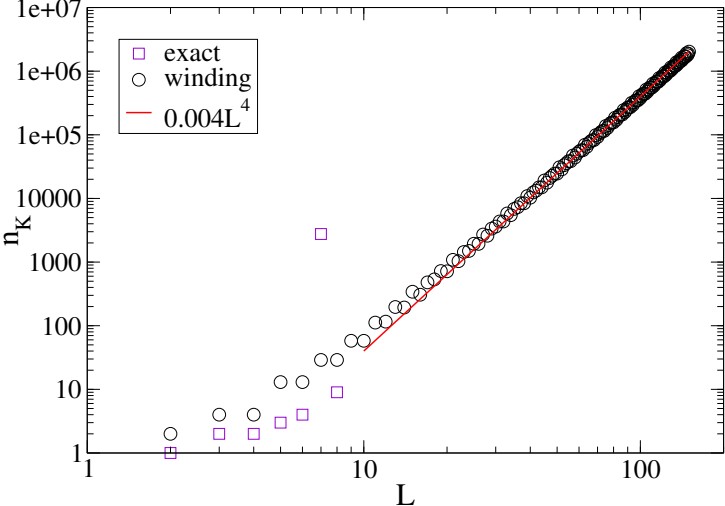

Figure 5: Number of Kempe sectors as a function of linear system size for the coloring problem (squares) and derived by assuming all configurations of winding numbers (circles). They form a finite fraction of the total number of configurations.

We have numerically iterated the dynamical transformations, Eqs. (24)-(26), from the initial configurations until the sectors close. In Fig. 5, we show the number of Kempe sectors of winding numbers as a function of system size $L$. At small $L$, we can compare with that of table 1:

| $L$ | 2 | 3 | 4 | 5 | 6 | 7 | 8 |
|---|---|---|---|---|---|---|---|
| $n_K$ (exact) | 1 | 2 | 2 | 3 | 4 | 2761 | 9 |
| $n_K$ (winding) | 2 | 4 | 4 | 13 | 13 | 29 | 29 |

The differences between the two lines have two origins. First, some of the sets of integers $(a, b, c)$ that do not exist in the coloring problem form additional Kempe sectors by themselves. Second, for special $L$ ($L = 7$), a few $(a, b, c)$ classes are themselves split into subsectors (as discussed in section 2), giving additional Kempe sectors.

We now have access to much larger system sizes and we can study the number of sectors of the winding number configurations. At large $L$, $n_K$ converges to $0.004L^4$ (solid line), which is a finite fraction of all configurations. We emphasize that it is not the number of Kempe sectors of the original problem and we do not know how many configurations among them are unphysical. A first point, however, is that the winding number configurations themselves form a large number of disconnected sectors.

---

[3]This is indeed verified for the sizes available, $L = 1, \ldots, 8$. All winding numbers with $n^2 \leq 5$ are realized for $L = 7, 8$ (but not for $L < 7$).

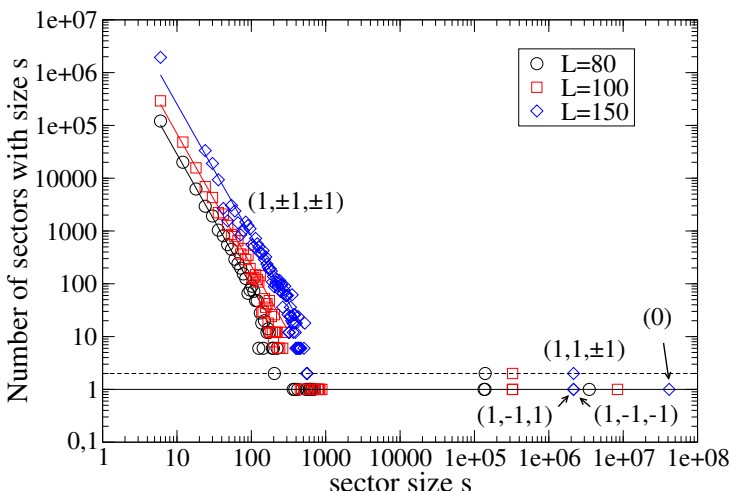

Figure 6: Distribution of sizes of Kempe sectors (log-log scale). It has a large number of small sectors (smaller than $\sim L$) and five sectors around macroscopic values (the dashed line indicates two sectors). The five larger sectors are labeled by the invariants found in section 4 (shown for $L = 150$, the two with $(1, -1, \pm 1)$ are almost degenerate).

In Fig. 6, we show the distribution of the number $s$ of configurations within the Kempe sectors. It has three peaks: one peak around zero with a small number of states in each sector, and two peaks with a macroscopic number of states (a fraction of $L^4$) split into five sectors (among them two have exactly the same number of states, and two others differ by a few percents). The distribution around zero is close to a power law (shown with lines in Fig. 6), $\sim s^{-2.5}$. Given the divergence of its integral at zero, the total number of sectors $n_K$ in Fig. 5, is dominated by the smallest sectors, those with only six states.

In summary, we observe two types of Kempe sectors, those with $s \ll L^4$ and five large sectors. We will now explain that the sectors can be labeled by three invariants that are polynomials of the winding numbers (section 4), and that the small sectors result from the steric constraints imposed by Eqs. (18) (section 5).

## 4 Stable invariants

In this section, we show some conservation laws, *i.e.* explicit functions of $(\boldsymbol{a}, \boldsymbol{b}, \boldsymbol{c})$ that are conserved by the integer dynamics and take different values in the different sectors. They are stable invariants, valid for all system sizes.

### 4.1 Conservation of the parity of the number of line crossings

We define

$$\boldsymbol{a} \times \boldsymbol{b} = a_x b_y - a_y b_x, \tag{33}$$

which is a one-component integer (the other components, such as $a_z b_x - a_x b_z = -(a_x + a_y)b_x + a_x(b_x + b_y) = a_x b_y - a_y b_x$, are equal to it). It counts the number of geometrical (signed) crossings of $a$-type loops with $b$-type loops.

Its symmetric version is

$$\chi = \frac{1}{3}(a \times b + b \times c + c \times a), \tag{34}$$

since, $a \times b = b \times c = c \times a$ results from $a + b + c = 0$. So, for instance, if there are no A-B winding loops, $c = 0$ and $\chi = 0$: B-C and A-C winding loops (if they exist) are parallel in the geometrical sense.

In the transformation Eq. (24),

$$a' \times b' = a \times b + 2k\hat{a} \times b \tag{35}$$

has its parity conserved. The same argument applies for the two other transformations Eq. (25) and (26). Therefore, the quantity

$$I_2^a = \chi \bmod 2 \tag{36}$$

is conserved by the dynamics. $I_2^a$ defines two sectors, characterized by an even or odd number of geometrical crossings of the winding loops. The sector of even parity is *connected* and the proof (given in Appendix A) consists of showing that all configurations are connected to a target configuration. The sector of odd parity is *not* connected and we provide below further invariants in the odd sector.

### 4.1.1 Relationship with odd/even chirality invariant

We now explain why it is the same invariant as the parity of the chirality, previously defined [17]. We define the chirality at a vertex by choosing an orientation and count $+1/2$ for an ABC ordering and $-1/2$ for ACB (see Fig. 7, the same definition applies for black and white vertices). The chirality of a color configuration is the sum over the $2L^2$ vertices and is an odd or even integer number between $-L^2$ and $+L^2$. Its parity is conserved by the dynamics [17].

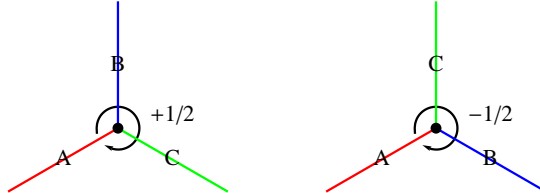

Figure 7: Definition of the chirality of a vertex.

On the other hand, consider the winding loops of B-C type (called $a$-type, and represented by black lines in Fig. 8) and winding loops of C-A type (called $b$-type, and represented by dashed lines). If they cross, they cross in edges colored C. There are three situations, shown in Fig. 8: when the chirality of the two vertices adds up to zero (left figure), there is no crossing at all. When the chirality is $+1$ (middle figure), the loops cross and the crossing has the $+$ sign. When the chirality is $-1$ (right figure), the sign is $-1$. Now if we consider all loops (winding and nonwinding) they cross (or anti-cross) in one third of all edges. A third of edges connects two vertices only once, so that summing over the C edges is equivalent to summing over all vertices. The total chirality is, therefore, equal to the number of signed crossings. Winding and nonwinding loops, on one hand, nonwinding and nonwinding loops, on the other hand, have zero signed crossings; so that the total chirality is the number of signed crossings of the winding loops. The parity of the chirality and the parity of the number of crossings are the same invariant.

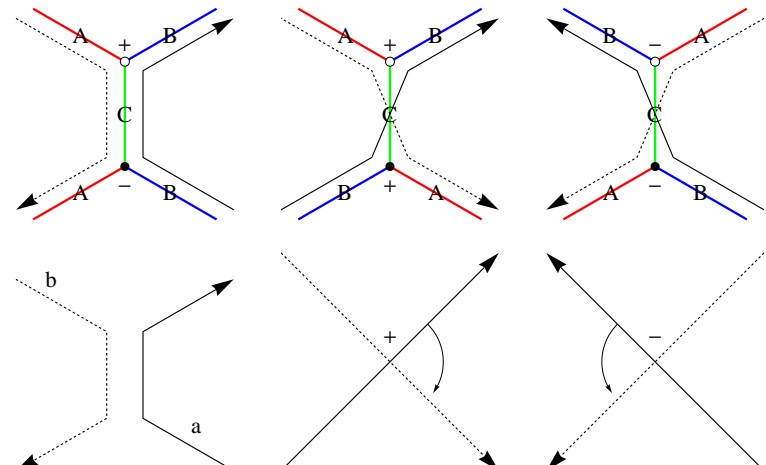

Figure 8: Correspondence between chirality and signed crossing of loops: $a$ (black) and $b$ (dashed) type loops cross at a C site. Left: They anti-cross when the chirality of the black and white vertex sums to zero. Middle and right: They cross with a $\pm$ sign when the chirality is $\pm 1$.

### 4.2 Conservation of the norm modulo 4 for odd number of crossings

Since the even sector is connected, we now restrict the configurations to those which have an odd number of crossings and define

$$
\begin{align}
\boldsymbol{a} \cdot \boldsymbol{b} &= a_x b_x + a_y b_y + a_z b_z \tag{37} \\
&= a_x b_x + a_y b_y + (a_x + a_y)(b_x + b_y) \tag{38} \\
&= 2a_x b_x + 2a_y b_y + a_x b_y + a_y b_x. \tag{39}
\end{align}
$$

$\boldsymbol{a} \cdot \boldsymbol{b}$ is necessarily an odd integer, because in Eq. (39), $a_x b_y$ and $a_y b_x$ cannot be simultaneously odd or simultaneously even when the number of crossings, $a_x b_y - b_x a_y$ is odd: one is odd, one is even. With $\boldsymbol{a} \cdot \boldsymbol{b}$ odd, $\boldsymbol{a} \cdot \boldsymbol{b} \equiv 1 \pmod 4$ or $\boldsymbol{a} \cdot \boldsymbol{b} \equiv 3 \pmod 4$, which, following a standard convention, we note $-1 \equiv 3 \pmod 4$. So that $\boldsymbol{a} \cdot \boldsymbol{b} \equiv \pm 1 \pmod 4$. Now, if we consider $\boldsymbol{a} \cdot \boldsymbol{c}$, we have

$$
\boldsymbol{a} \cdot \boldsymbol{c} = -\boldsymbol{a} \cdot \boldsymbol{b} - 2(a_x^2 + a_y^2 + a_x a_y). \tag{40}
$$

But $a_x^2 + a_y^2 + a_x a_y$ is odd, because $a_x$, or $a_y$, or both are odd. Because $\boldsymbol{a} \cdot \boldsymbol{b}$ is odd, we get a sign change and $\boldsymbol{a} \cdot \boldsymbol{c} \equiv \boldsymbol{a} \cdot \boldsymbol{b} \pmod 4$. Similarly, $\boldsymbol{a} \cdot \boldsymbol{b} \equiv \boldsymbol{b} \cdot \boldsymbol{c} \pmod 4$. Hence we can define a symmetric version,

$$
I_2 = \frac{1}{3}(\boldsymbol{a} \cdot \boldsymbol{b} + \boldsymbol{b} \cdot \boldsymbol{c} + \boldsymbol{c} \cdot \boldsymbol{a}) \bmod 4, \tag{41}
$$

where the three terms are, in fact, equal, giving $I_2 = \pm 1$ (recall $-1 \equiv 3 \pmod 4$). It can also be written by simple rearrangements,

$$
I_2 = -\frac{1}{6}(|\boldsymbol{a}|^2 + |\boldsymbol{b}|^2 + |\boldsymbol{c}|^2) \bmod 4 = -n^2 \bmod 4. \tag{42}
$$

In this form, we see that $I_2$ is nothing but the norm squared of the total windings, modulo 4.

After a transformation along $a$, with a factor $k$ [Eq. 24], we get,

$$
\begin{align}
I_2' &= \frac{1}{3}(\boldsymbol{a}' \cdot \boldsymbol{b}' + \boldsymbol{b}' \cdot \boldsymbol{c}' + \boldsymbol{c}' \cdot \boldsymbol{a}') \bmod 4 \tag{43} \\
&= I_2 - k(k + \gcd(a_x, a_y))\hat{a}^2 \bmod 4. \tag{44}
\end{align}
$$

We now show that the variation vanishes for all $k$. First, since both components $a_x$, $a_y$ cannot be simultaneously even in the odd sector, $\gcd(a_x, a_y)$ is odd; and $k(k + \gcd(a_x, a_y))$ is even. $\hat{a}^2$ results from the definition of the scalar product, Eq. (37): $\hat{a}^2 = 2(a_x^2 + a_y^2 + a_x a_y)$ is even. Therefore, $k(k + \gcd(a_x, a_y))\hat{a}^2$ is a multiple of 4 and $I_2' = I_2$. The same argument applies for the other two transformations by symmetry. $I_2$ is therefore a conserved quantity and, since it takes two values, $I_2 = \pm 1$, it defines two sectors, within the odd sector.

## 4.3 Conservation of the sign for odd number of crossings

The third invariant is associated to the observation that the dynamics cannot reverse the sign: a given configuration $(a, b, c)$ and its *antipodal* configuration $(-a, -b, -c)$, when it exists, belong to two different sectors. They cannot be distinguished by $I_2^a$ or $I_2$ which are both of even degree and have the same value for both configurations, so it must be proven by a different invariant. We define two quantities,

$$I^{\pm} = \frac{1}{2}[I_6 + I_5^{\pm}] \bmod 4, \tag{45}$$

where the sign $\pm$ refers to $I_2 = \pm 1$, and

$$\begin{aligned} I_6 &= a_x a_y a_z b_x b_y b_z + b_x b_y b_z c_x c_y c_z + c_x c_y c_z a_x a_y a_z, \tag{46} \\ I_5^{\pm} &= (a_x \pm a_y)(a_y \pm a_z)(a_z \pm a_x)\chi \\ &+ (b_x \pm b_y)(b_y \pm b_z)(b_z \pm b_x)\chi \\ &+ (c_x \pm c_y)(c_y \pm c_z)(c_z \pm c_x)\chi, \tag{47} \end{aligned}$$

where $\chi = \frac{1}{3}(a \times b + b \times c + c \times a)$ [Eq. (34)] is odd. We will show that $I^{\pm}$ are indeed conserved by the dynamics in their respective sectors $I_2 = \pm 1$ and allow to distinguish a configuration from its antipodal configuration.

In the odd sector, one and only one component in each vector $a$, $b$ and $c$ is even (see Eq. (49) below). $I_6$, which contains the product of two such components, is therefore a multiple of 4, *i.e.* $\frac{1}{2}I_6 \equiv 0, 2 \pmod 4$. Similarly, $I_5^{\pm}$ is an odd sum of products of four terms. In each product, one term and only one is even, so that $\frac{1}{2}I_5^{\pm}$ is odd, hence $\frac{1}{2}I_5^{\pm} = \pm 1 \pmod 4$, and therefore $I^{\pm} = \pm 1$. It takes opposite signs for $(a, b, c)$ and $(-a, -b, -c)$. Indeed, the first term $\frac{1}{2}I_6$ is unchanged, but the second term, which carries the sign, is reversed, so that $I^{\pm} \to -I^{\pm}$. We have shown numerically that $I^{\pm}$ are conserved. They thus label the two "antipodal" sectors. Here we give the explicit proof that $I^+$ is an invariant. We first show that it is invariant in permutations, *e.g.*

$$(a, b, c) \to (-b, -a, -c). \tag{48}$$

$I_6$ is symmetric in the permutation of $a, b, c$ and has even degree so the change of sign has no effect. The term $(a_x \pm a_y)(a_y \pm a_z)(a_z \pm a_x)$ becomes $-(b_x \pm b_y)(b_y \pm b_z)(b_z \pm b_x)$, but $\chi = a \times b \to (-b) \times (-a) = -\chi$ so the product is unchanged.

We now consider the transformations $T_i(k)$ ($i = a, b, c$), with $k = \pm 1$, without loss of generality. Since we have the invariance in permutations, we will always permute the configurations to an equivalent configuration identified by a value of $(a, b, c) \bmod 2$. We choose, for instance,

$$(a, b, c) \bmod 2 = (1, 0, 1, 0, 1, 1, 1, 1, 0), \tag{49}$$

which is one of the six possible configurations for odd $a \times b$, the other ones being obtained by permutations of the three vectors $(1, 0, 1)$, $(0, 1, 1)$, $(1, 1, 0)$. It has three even components $a_y$, $b_x$ and $c_z$, so that $a_x b_y$ is odd, $a_y b_x$ even, and the difference $a \times b$ is, indeed, odd.

For these configurations, $I^{\pm}$ simplifies and we give the simplified expression of $I^+$, *i.e* when $I_2 = +1$.

In order to simplify $I_6$, we use the fact that the product of an odd by an even number modulo 4 equals the even number: $(2p+1) \times 2n \equiv 2n(\text{mod } 4)$. We thus have,

$$\frac{1}{2}a_x a_y a_z b_x b_y b_z(\text{mod } 4) = \frac{1}{2}a_y b_x(\text{mod } 4), \tag{50}$$

because $a_x a_z b_y b_z$ is odd and $\frac{1}{2}a_y b_x$ even, so that

$$\frac{1}{2}I_6(\text{mod } 4) = \frac{1}{2}(a_y b_x + b_x c_z + c_z a_y)(\text{mod } 4). \tag{51}$$

We show that the sum of the last two terms vanish. If $c_z$, which is even, is a multiple of 4, then $\frac{1}{2}c_z(b_x + a_y) \equiv 0(\text{mod } 4)$ because $b_x + a_y$ is also even. If $c_z \equiv 2(\text{mod } 4)$, we get $c_z = a_x + a_y + b_x + b_y \equiv 2(\text{mod } 4)$. Consider $a_x + b_y$. By using the definition $I_2 = \boldsymbol{a} \cdot \boldsymbol{b}(\text{mod } 4)$ and Eq. (39), we get $I_2 = a_x b_y(\text{mod } 4) = 1$. Since $a_x b_y = (2n+1)(2p+1) = 1+2(n+p)(\text{mod } 4)$, we must have $n + p$ even. As a consequence, $a_x + b_y = 2 + 2(n + p) \equiv 2(\text{mod } 4)$, hence $b_x + a_y \equiv 0(\text{mod } 4)$. Therefore, in this case too, $\frac{1}{2}c_z(b_x + a_y) \equiv 0(\text{mod } 4)$ and

$$\frac{1}{2}I_6(\text{mod } 4) = \frac{1}{2}a_y b_x(\text{mod } 4). \tag{52}$$

We now consider $I_5^+$ [Eq. (47)], and replace $(a_x + a_y)$ by $-a_z$ etc.,

$$I_5^+ = -(a_x a_y a_z + b_x b_y b_z + c_x c_y c_z)\chi. \tag{53}$$

We note that,
$$3a_x a_y a_z = a_x^3 + a_y^3 + a_z^3. \tag{54}$$

Since $(2n+1)^3 \equiv 2n+1(\text{mod } 8)$ and $(2n)^3 \equiv 0(\text{mod } 8)$, and since only $a_y$ is even, Eq. (54) reduces, modulo 8, to

$$a_x^3 + a_z^3(\text{mod } 8) = a_x + a_z(\text{mod } 8) = -a_y(\text{mod } 8). \tag{55}$$

Now since $3 \times 2n \equiv -2n(\text{mod } 8)$, we get $a_x a_y a_z(\text{mod } 8) \equiv a_y(\text{mod } 8)$. We thus obtain,

$$\frac{1}{2}I_5^+(\text{mod } 4) = -\frac{1}{2}(a_y + b_x + c_z)\chi(\text{mod } 4). \tag{56}$$

Consider $\chi(\text{mod } 4) = a_x b_y - a_y b_x(\text{mod } 4) = I_2 = +1$ for the current value of the parity. Note that $\chi$ is globally *not* conserved modulo 4 (only modulo 2) because it changes sign in permutations. We can replace $\chi \equiv 1(\text{mod } 4)$ and combine the two equations,

$$I^+ = \frac{1}{2}(a_y b_x - (a_y + b_x + c_z))\text{mod } 4, \tag{57}$$

which can be used provided that the configuration has the parity given by Eq. (49). Since, however, it breaks the symmetry between the three types of loops, we have to consider the three types of transformations in turn.

### 4.3.1 Invariance in the insertion of $a$-type loops

The transformation $T_a(k = \pm 1)$ is combined with a permutation, $P_a.T_a(k = \pm 1)$, in order to conserve the correct parity pattern of Eq. (49). Applying this transformation to a general configuration $(\boldsymbol{a}, \boldsymbol{b}, \boldsymbol{c})$, we obtain

$$(\boldsymbol{a}', \boldsymbol{b}', \boldsymbol{c}') = (\boldsymbol{a} - 2(g+k)\hat{\boldsymbol{a}}, \boldsymbol{b} + (g+k)\hat{\boldsymbol{a}}, \boldsymbol{c} + (g+k)\hat{\boldsymbol{a}}), \tag{58}$$

where $g = \gcd(a_x, a_y)$. We see, indeed, that

$$(\boldsymbol{a'}, \boldsymbol{b'}, \boldsymbol{c'}) \bmod 2 = (\boldsymbol{a}, \boldsymbol{b}, \boldsymbol{c}) \bmod 2, \tag{59}$$

because $g$ is odd in the odd sector and $k = \pm 1$. By using the transformation Eq. (58) and the form [Eq. (57)] of the invariant, we obtain,

$$\begin{aligned}
(I^+)'_a &= \frac{1}{2}(a'_y b'_x - (a'_y + b'_x + c'_z)) \bmod 4 \tag{60} \\
&= I^+ + \frac{1}{2}(g+k)\hat{a}_y(g\hat{a}_x + 3) \bmod 4, \tag{61}
\end{aligned}$$

after removing two multiples of 4. Now, $\frac{1}{2}(g+k)$ is an integer, $\hat{a}_y$ is even according to Eq. (49), $g\hat{a}_x$ is odd (both are odd), so that $(g\hat{a}_x + 3)$ is even and the product is a multiple of 4. Therefore $(I^+)'_a = I^+$.

### 4.3.2 Invariance in the insertion of $b$-type loops

Consider the action of the parity-conserving transformation $P_b.T_b(k = \pm 1)$,

$$(\boldsymbol{a'}, \boldsymbol{b'}, \boldsymbol{c'}) = (\boldsymbol{a} + (g+k)\hat{\boldsymbol{b}}, \boldsymbol{b} - 2(g+k)\hat{\boldsymbol{b}}, \boldsymbol{c} + (g+k)\hat{\boldsymbol{b}}), \tag{62}$$

where $g = \gcd(b_x, b_y)$. The invariant is similar,

$$(I^+)'_b = I^+ + \frac{1}{2}(g+k)\hat{b}_x(g\hat{b}_y + 3) \bmod 4, \tag{63}$$

where $(g+k)$, $\hat{b}_x$ and $(g\hat{b}_y + 3)$ are all even. The variation therefore vanishes, $(I^+)'_b = I^+$.

### 4.3.3 Invariance in the insertion of $c$-type loops

Last, consider similarly the action of $P_c.T_c(k = \pm 1)$,

$$(\boldsymbol{a'}, \boldsymbol{b'}, \boldsymbol{c'}) = (\boldsymbol{a} + (g+k)\hat{\boldsymbol{c}}, \boldsymbol{b} + (g+k)\hat{\boldsymbol{c}}, \boldsymbol{c} - 2(g+k)\hat{\boldsymbol{c}}), \tag{64}$$

where $g = \gcd(c_x, c_y)$. The invariant becomes,

$$(I^+)'_c = I^+ + \frac{1}{2}(g+k)(a_y\hat{c}_x + b_x\hat{c}_y + (g+k)\hat{c}_x\hat{c}_y + 3\hat{c}_z) \bmod 4.$$

The terms in brackets are even, because $(g+k)$ is even, and $a_y$, $b_x$, and $c_z$ are the three even components. If $\frac{g+k}{2}$ is even, then the variation is a multiple of 4 and vanishes. Consider $\frac{g+k}{2}$ odd, in the rest of the proof. By using, again, $(2p+1) \times 2n \equiv 2n \pmod 4$, we get

$$\begin{aligned}
(I^+)'_c &= I^+ + (a_y\hat{c}_x + b_x\hat{c}_y + (g+k)\hat{c}_x\hat{c}_y + 3\hat{c}_z) \bmod 4 \\
&= I^+ + (a_y + b_x + (g+k) - \hat{c}_z) \bmod 4,
\end{aligned}$$

where, in the second line, we have eliminated the odd factors multiplied by even numbers. Note that $g + k = 4n + 2 \equiv 2 \pmod 4$. Note also that $c_z = g\hat{c}_z = (4n+1)\hat{c}_z$ where $\hat{c}_z$ is an integer by definition, implying $\hat{c}_z \equiv c_z \pmod 4$. We now replace $\hat{c}_z$ by $c_z = a_x + a_y + b_x + b_y$,

$$(I^+)'_c = I^+ + (2 - a_x - b_y) \bmod 4.$$

Since $a_x + b_y \equiv 2 \pmod 4$ (see above), we have therefore $(I^+)'_c = I^+$.

Table 2: Examples of configurations of winding numbers $(a, b, c)$ with the smallest norm $n^2$, in the Kempe sectors labeled by $(I_2^a, I_2, I^{\pm})$, for $L \bmod 3 = 0, 1, 2$ (from top to bottom). One can check that $a + b + c = 0$, $a_x + a_y + a_z = 0$, $a - b \equiv L \pmod 3$ etc.

| $I_2^a$ | $I_2$ | $I^{\pm}$ | $(a, b, c)$ | $n^2$ |
|---|---|---|---:|---|
| 0 | - | - | 0 | 0 |
| 1 | 1 | $\pm 1$ | $\pm(2, -1, -1, -1, -1, 2, -1, 2, -1)$ | 3 |
| 1 | -1 | $\pm 1$ | $\pm(3, -1, -2, -3, 2, 1, 0, -1, 1)$ | 5 |

| $I_2^a$ | $I_2$ | $I^{\pm}$ | $(a, b, c)$ | $n^2$ |
|---|---|---|---:|---|
| 0 | - | - | $(0, 0, 0, -1, -1, 2, 1, 1, -2)$ | 2 |
| 1 | 1 | $\pm 1$ | $\pm(-3, 2, 1, 2, 1, -3, 1, -3, 2)$ | 7 |
| 1 | -1 | $-1$ | $(-1, 0, 1, 1, -1, 0, 0, 1, -1)$ | 1 |
| 1 | -1 | $1$ | $(-3, 4, -1, 5, -3, -2, -2, -1, 3)$ | 13 |

| $I_2^a$ | $I_2$ | $I^{\pm}$ | $(a, b, c)$ | $n^2$ |
|---|---|---|---:|---|
| 0 | - | - | $(0, 0, 0, 1, 1, -2, -1, -1, 2)$ | 2 |
| 1 | 1 | $\pm 1$ | $\pm(-3, 1, 2, 4, -1, -3, -1, 0, 1)$ | 7 |
| 1 | -1 | $1$ | $(1, 0, -1, -1, 1, 0, 0, -1, 1)$ | 1 |
| 1 | -1 | $-1$ | $(-1, 1, 0, 3, 2, -5, -2, -3, 5)$ | 13 |

## 4.4 Number of sectors from the invariants

The three invariants $I_2^a$, $I_2$ and $I^{\pm}$ allow to label the Kempe sectors. We have $I_2^a = 0$ for the even sector which is connected, and $(I_2^a, I_2, I^{\pm}) = (1, \pm 1, \pm 1)$ for the odd sectors. We thus have five sectors that define disconnected configurations.

In table 2, we give examples of the winding numbers with smallest norm $n^2$ [as defined by Eq. (17)], to show that each sector is, indeed, realized (provided that $L$ is large enough). We note that, when $L$ is not a multiple of three, antipodal sectors do not always exist, but different sectors of $I^{\pm}$ do exist. We also show the labels of the five large sectors in Fig. 6.

Since we have enlarged the number of winding configurations compared with the coloring problem, we have checked that all the values of the invariants are also realized in the coloring problem (see table 1).

There are still many disconnected sectors labeled by the same invariants (see Figs. 5 and 6) and this point is studied in section 5.

## 4.5 Symmetries

The two large sectors labeled by $(1, 1, \pm 1)$ have the same size, because they are the chiral image of each other in mirror plane symmetries $P$. For a given configuration of winding numbers, there is always another configuration obtained by exchanging two axis among the $x$, $y$ or $z$ axis. When $I_2 = 1$, we see that $I_5^+$ in Eq. (47) is reversed in an exchange of axis (because $\chi$ takes a minus sign). Hence $I^+ \to -I^+$ and the two mirror configurations are in separate sectors. Note, indeed, that there are special states $(a, b, c)$ with $a_y = a_z$, $b_y = c_z$, $b_z = c_y$, $b_x = c_x$ (see the example in the top table of Table 2), which a mirror plane $yz$ transforms effectively according to $(a, b, c) \to (a, c, b)$. An allowed permutation gives in turn $(a, c, b) \to (-a, -b, -c)$ which is the opposite of the original state. The antipodal transformation $T$, therefore, results in a lattice symmetry operation when $I_2 = 1$. As a consequence, the two sectors must have

the same size. This symmetry implies an equal number of homotopy classes in the two sectors (see the degeneracy in Fig. 6), and also an equal number of color configurations (see the multiplicity in table 1). Since a configuration in one of these sectors and its mirror image are not connected, the dynamical evolution remains chiral.

The two other sectors $(1, -1, \pm1)$ are nondegenerate, however. When $I_2 = -1$, a configuration and its mirror image are in the same sector: $I_5^- \to I_5^-$ [Eq. (47)] ($\chi$ cancels the sign of say $a_x - a_y$), hence $I^-$ is invariant in mirror planes. In this case, the antipodal transformation is not a symmetry. Note that, for a given $(a, b, c)$, $(-a, -b, -c)$ may violate the constraints (see Eqs. (14)-(16)), so that the asymmetry of the constraints implies that of the two sectors $(1, -1, \pm1)$.

The dynamics conserves $I^+ = \pm1$ which distinguishes two degenerate chiral sectors, and $I^- = \pm1$ that defines two antipodal achiral nondegenerate sectors.

# 5   Steric constraints, dilute limit

We argue that all the remaining disconnected sectors (see Fig. 6), which do share the same invariants $(I_2^a, I_2, I^\pm)$, cannot be distinguished by any other invariant. The remaining obstruction is not a property of the transformations, but is of steric origin.

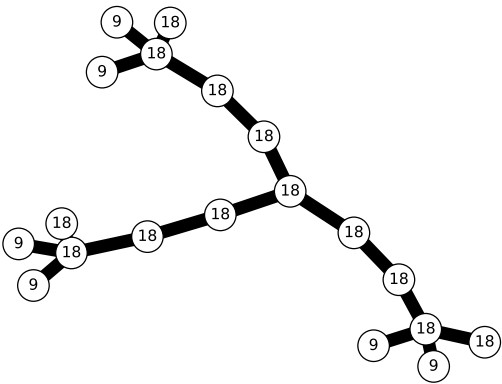

Figure 9: Example of a Kempe sector with $(I_2^a, I_2, I^\pm) = (1, -1, 1)$ for $L = 18$, represented as a graph: nodes (winding number configurations $(a, b, c)$, up to permutations) are connected by the dynamics. Indicated is the minimal size $L$ for the configuration to fulfill the constraints, thus showing that the paths between $L = 9$ configurations are cut, giving $1 \to 6$ isolated sectors by steric obstruction.

We show this by allowing larger sizes at fixed and small winding numbers. As earlier, we construct the Kempe sectors for a given size $L$. They are each characterized by a set of winding numbers up to $L$. The same set of winding numbers can be considered at larger sizes $L_1 \geq L$ (provided that $L_1 \equiv L \pmod 3$). We can now redo the construction of the sectors starting with the same winding numbers. In the dynamics, larger winding numbers are generated and we test whether they open new paths between the original winding configurations. In Fig. 9, we give an example of the graph of a Kempe sector. The nodes are winding loop configurations $(a, b, c)$ (up to permutations), connected by edges when a dynamical transformation between them exists. The number indicated in each node is the minimal size $L$ for the winding numbers to fulfill the size constraints. We see then clearly that the winding numbers existing for $L = 9$, for instance, get reconnected through winding number configurations that exist only for $L_1 \geq L$. The path between them implies configurations with higher winding numbers.

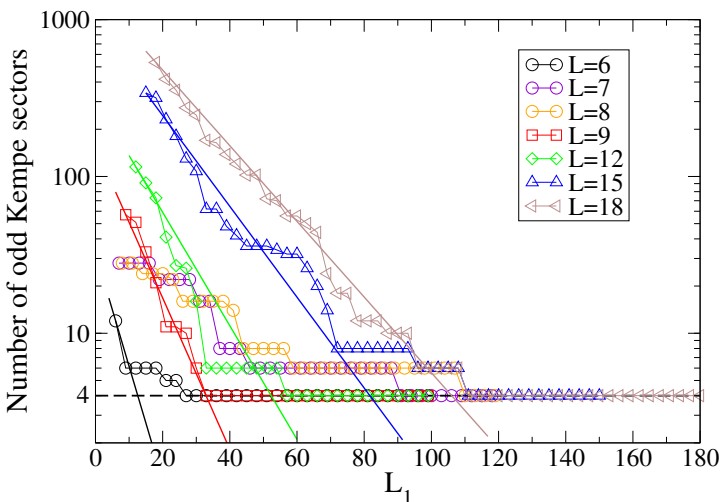

Figure 10: Multiscale reconnection of the odd Kempe sectors at size $L$ when winding numbers up to $L_1$ are allowed. By increasing $L_1$, the number of sectors decrease, showing reconnections, down to the four (dashed line) odd sectors characterized by the invariants. Decays as $\sim \exp(-L_1/L)$ are shown for comparison (solid lines).

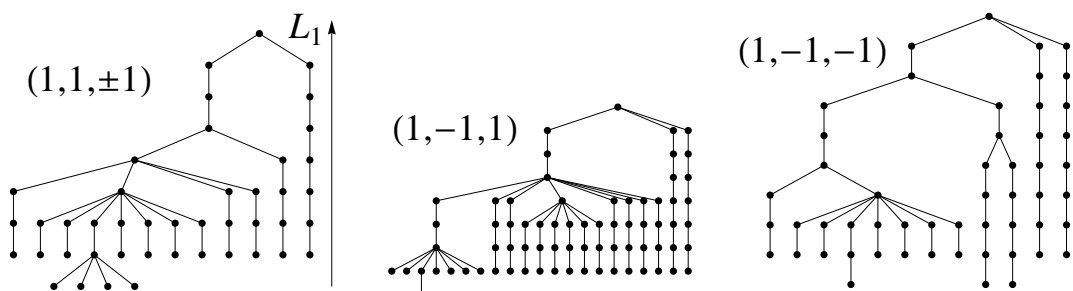

Figure 11: Configuration space of low-winding numbers as four hierarchical trees (one for each invariant $(1, \pm1, \pm1)$, two are identical by symmetry). Each point represents a Kempe sector. The level $n \geq 0$, from bottom up corresponds to the system size, $L_1 = 6 + 3n$, *i.e.* higher winding numbers allowed. Only the lowest two levels $n = 0, 1$ include *all* the sectors. For $n \geq 2$, additional sectors are not shown.

In Fig. 10, we see that, as the size $L_1$ increases, the number of sectors constructed for size $L$ decrease, so that new connections indeed appear. Moreover, in all cases considered, all isolated odd sectors merge into four sectors (characterized by the invariants of section 4) for large enough $L_1$ (dashed line), *i.e.* when the system is diluted, $|a|, |b|, |c| \ll L_1$. The fact that the process converges to four means that there are no other invariants (it cannot go lower than 4 because of the distinct invariants). There exist paths between the winding loop configurations, provided that the size is large enough. In other words, the disconnection at size $L$, beyond that due to the invariants, is a steric obstruction. This is illustrated in Fig. 11 where all the odd Kempe sectors at $L = 6, 9$ are represented by points in the first two bottom levels. The level $n$ corresponds to the size $L_1 = 6 + 3n$ ($n \geq 0$). The lower levels get reconnected at higher levels (through higher winding numbers). Note that for $n \geq 2$ not all the sectors are shown.

While this occurs for the winding number configurations, the reconnection could take place through unphysical configurations (see section 3.3) and may be unphysical for the coloring problem. As noted earlier, the configurations in the dilute limit are not expected to be constrained by the discreteness of the lattice, so that we expect all sets of winding numbers in this limit.

The restoration of ergodicity (equilibration among the sterically-obstructed sectors) is described in Fig. 10 by an exponential decrease $\sim \exp(-L_1/\xi)$, where $\xi = L$ fits approximately (solid lines). It means that the path between any two loop configurations with low winding numbers implies configurations with higher winding numbers of order a few $\xi$ (which we could think of as higher energy states, if an energy were to be defined). Since there is a distribution of higher winding numbers, the equilibration is thus a multiscale process. The image is that of a space of configurations consisting of hierarchical trees (Fig. 11). A physical model of equilibration of configurations with low winding numbers may thus involve the ultrametric distance between sectors: the dynamics is expected to be slow [19].

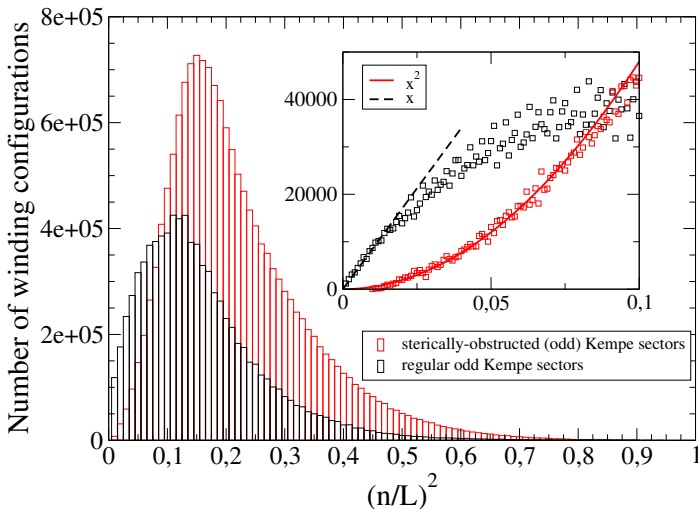

Figure 12: Probability distribution of norms, $n^2$ in winding loop configurations, for regular odd sectors and sterically-obstructed (odd) Kempe sectors ($L = 150$). Small norms (see inset) are absent from the latter: linear (dashed line) and quadratic (solid line) fits are added.

In order to confirm the connectedness of the space of configurations in the dilute limit for larger $L$, we can study the distribution of winding numbers. In Fig. 12, we give the probability distribution function of the norm squared, $n^2 = \frac{1}{6}(|\boldsymbol{a}|^2 + |\boldsymbol{b}|^2 + |\boldsymbol{c}|^2)$, over all configurations for $L = 150$, by distinguishing configurations of regular or sterically-obstructed sectors. Overall, it is apparent that a randomly-chosen odd configuration has more chance to belong to a sterically-obstructed sector. The probability for a configuration to be odd is 0.37 (0.63 to be even), and within the odd sector, the probability is 0.65 to belong to a sterically-obstructed odd sector (0.35 to a regular odd sector). Sterically-obstructed sectors have all norms above a small size-dependent threshold. In the limit of small $(\frac{n}{L})^2 \ll 1$ (inset of Fig. 12), the number of such sectors is vanishingly small, in $\sim (\frac{n}{L})^4$ (solid line), whereas the number of regular sectors varies like $(\frac{n}{L})^2$ (dashed line). A randomly-chosen configuration therefore belongs to a regular sector with a probability $1 - \alpha(\frac{n}{L})^2 \to 1$ in the limit $n \ll L$, and is hence connected to all other configurations with the same label. This confirms the existence of a proper ergodic dilute limit where the only obstruction to ergodicity is that described by the invariants. On

the other hand, the probability to belong to a sterically-obstructed sector increases steadily as $n$ increases, in agreement with the picture that there is less and less space when $n \to L$ and that steric obstruction is more and more prominent.

## 6 Conclusion

We have obtained a classification of the Kempe sectors on the regular hexagonal lattice for generic sizes, based on the loop winding numbers. We have found a set of five sectors labeled by three invariant parities that are polynomials of the winding numbers. The first invariant has a simple geometrical interpretation, this is the parity of the number of crossings of the winding loops. When this number is even, the sector is connected. When it is odd, it splits into four sectors, two achiral and two chiral. The two chiral sectors have the same number of classes (degenerate) as they are image of each other by lattice mirror symmetries. Thus, there are configurations that break mirror symmetries, that the dynamics cannot restore (contrary to spontaneous symmetry breaking). The two achiral sectors do not break the mirror symmetry and are nondegenerate. They are "antipodal", a symmetry that is broken by the "box" constraints. We assume that the invariants may be viewed geometrically as linking numbers of flux lines on the lattice, which is embedded (or immersed) into a 3d manifold.

At fixed $L$, there are many additional undistinguished sectors that share the same invariants. We have shown that they would be connected to the four odd sectors if there was enough space to accommodate higher winding numbers. For these additional sectors, the problem is therefore a steric obstruction, not a conservation law. We have thus argued that the set of three invariants is complete. This allowed to describe the configuration space by distinguishing dilute from more dense configurations. While dense configurations can be sterically-obstructed, a randomly-chosen configuration in the dilute limit $|a|, |b|, |c| \ll L$ (where space is available) belongs to one of the five sectors and can be connected to any other with the same label, with no further obstruction. Moreover, the reconnection dynamics takes place on hierarchical trees and involves a multiscale distribution of intermediate winding numbers. For this reason, the equilibration dynamics of the low-flux lines, up to the conservation laws, is expected to be "slow". This gives some perspective as to how an entanglement of flux lines could lead to slow dynamics [20]: here it results from the need of a distribution of higher-flux intermediate configurations.

## Acknowledgements

O.C. would like to thank P. Dehornoy and R. Bacher for their helpful guidance.

## A  Sector with even number of line crossings is connected

We consider here the winding number configurations $(a, b, c)$ with $a \times b$ *even* and show that they are all connected by the dynamics, Eqs. (24)–(26). We restrict the proof for $L$ a multiple of three.

Start from the "0-sector" defined by $a = b = c = (0, 0, 0)$, which exists only when $L$ is a multiple of three. The sector is even since $a \times b = 0$. It will be our target state. Create a pair

of $a$-type winding loops,

$$a = 2m \tag{65}$$
$$b = -m \tag{66}$$
$$c = -m, \tag{67}$$

where $m$ is an arbitrary vector. A second transformation adds a pair of $b$-type loops with the same $m$,

$$a = 3m \tag{68}$$
$$b = -3m \tag{69}$$
$$c = 0. \tag{70}$$

Now we can add a pair of $c$-type winding loops with a different $m_1$,

$$a = 3m - m_1 \tag{71}$$
$$b = -3m - m_1 \tag{72}$$
$$c = 2m_1, \tag{73}$$

and $a \times b = 6m_1 \times m$ is even (since the parity is invariant).

Conversely, consider $L$ a multiple of 3, and any $(a, b, c)$ representing an even configuration. When $a \times b = a_x b_y - a_y b_x$ is even, we have to inspect 10 out of 16 parity configurations for the four independent integers $(a_x, a_y, b_x, b_y)$.

(i) they are all odd. Then $c = -a - b$ has three even components. Define $m_1 = c/2$, it is an integer vector, so is $a + c/2 = (a - b)/2$. When $L$ is a multiple of 3, the three components of $b - a$ are multiples of 3, according to Eq. (16). We can therefore define an integer vector $m$ such that $3m = a + c/2$. Now the vectors $(a, b, c)$ have the form of Eqs. (71)-(73). Inverting all the transformations above, transforms this sector onto the 0-sector.

(ii) they are all even. Again we can define $m_1 = c/2$, and the same argument applies.

(iii) $a_x$ even, $b_y$ odd, and $a_y$ even, $b_x$ odd. Then $c = -a - b$ is odd. But $a$ is even, so define $m_1 = a/2$ and the same argument applies again up to a permutation.

(iv) $a_x$ even, $b_y$ odd, and $a_y$ odd, $b_x$ even. Then $c_x = -a_x - b_x$ is even, and $c_y = -a_y - b_y$ is even. Define $m_1 = c/2$ etc. The other cases are similar.

Therefore any set of winding numbers $(a, b, c)$ such that $a \times b$ is even can be cast into the form of Eqs. (71)-(73) with the definition of two vectors $m, m_1$ which by three transformations can be transformed onto the "0-sector". Therefore, any even configuration is connected to any other even configuration.

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
