# Peer review of "Invariants of winding-numbers and steric obstruction in dynamics of flux lines"

_SciPost Physics, doi:SciPost Phys. 7, 032 (2019)_

## Round 1 · Referee Report · Anonymous (Referee 1) · 2019-8-15

Report
Requested changes
1) p.1: "is therefore or particular interest" -- typo
2) p.4: "form self-avoiding closed loops" -- may it be worth to point out to the unfamiliar reader that the loops are fully packed?
3) Fig.1 and 2 are not legible in black and white printing, and this could perhaps be easily fixed to spare the need to print in colour
4) Eq.(25) contains a typo; the middle term on the right hand side should be b+2kb rather than b-2kb, I think
5) many equations are missing a punctuation at the end -- starting from the equation between Eq.(27) and (28), which should be followed by a full stop
6) p.10: "While in the continuum limit, we would" -- no comma?
7) p.15: "...cannot be... neither...nor" is a double negative, which I presume is not intended to be
Author: Olivier Cépas on 2019-08-23 [id 583]
(in reply to Report 1 on 2019-08-15)We would like to thank the referee warmly for his/her report and very careful reading. We have included all the well-needed requested changes, points 1 to 7.
Author: Olivier Cépas on 2019-08-23 [id 584]
(in reply to Report 2 on 2019-08-21)We would like to thank the referee warmly for his/her report and encouragements. We provide some very partial answers to the interesting questions raised.
(1) We have included this precision in a separate note to keep this discussion brief.
(2) This is an interesting point of view which may be useful for further developments or simplifications.
(3) The question is difficult and we do not know an immediate generalization of the invariants for g>1, at present. Such a lattice could be defined by using k-gons with k>6 and the winding number vectors (a,b,c) generalized to vectors with 2g coordinates. While axb and a.b are in vectorial form, suggesting they could still be invariants, it is not clear how the third invariant could be generalized. It would be, of course, particularly interesting if the number of invariants depends on g.
(4) We have not investigated this point further because some of the sectors that enter this number, 0.004L⁴, may not be physical, i.e. may not correspond to acceptable homotopy classes of the original coloring problem, as mentioned in the text. We do not know in particular if there remains a finite fraction of L⁴ or not, when the unphysical part is removed. A way perhaps to understand this point further would be to determine which sectors are physical and which are unphysical.
(5) is corrected.
(6) Thank you for raising this point too, especially as we are confused by scientific usage and opposite views from different native speakers. As both ways are fine with us, we leave this point with the editor.
(7) is corrected.

---

## Round 1 · Referee Report · Anonymous (Referee 2) · 2019-8-21

Strengths
Weaknesses
Report
(1) As noted by the authors, the model relates to kagome systems with constraints. It might be useful to further explicitly and very briefly note that the center of the links of the honeycomb lattice studied here form the sites of a kagome lattice. The 3-coloring problem as it relates to links on the honeycomb lattice maps becomes the 3-coloring problem of the vertices of the kagome lattice.
(2) There have been other frameworks for studying the (3-) coloring problems on the links of general graphs (including lattices such as the kagome). One may ascribe a (cubic) roots of unity to each link or vertex. For the 3-coloring problem on the kagome lattice, this relates to configurations in which cubic roots of unity are associated with individual lattice site such that for two neighboring lattice sites $x \neq y$ (or, equivalently (given that $x^3=y^3=1$), these satisfy $x^2+ xy+y^2 =0$). The allowed 3-colorings can be related to polynomials in the Grobner basis.
(3) How might the results qualitatively change if the lattice was not on a torus but rather on a manifold with genus number $g$? Would the sector multiplicities be changed by such a change of topology?
(4) Is there is an intuitive rationale for the observed uniform fraction scaling for the winding number $n_{K}$ such as that of $0.004L^{4}$ found by the authors for $L \gg 1$?
Very more minor items of style:
(5) The sentence "Note that this symmetry is true in terms of number of homotopy classes (Fig. 6), but also in terms of number of color configurations (table 1)" did not read very well at this end.
(6) The word "dynamics" is plural. In the paper, the authors very consistently used it in singular form. One may change that to plural throughout the text.
(7) A colon may appear immediately after a word instead of a space (e.g., "First example: permutation of colors" instead of "First example : permutation of colors" and other similar subtitles and text).
I must apologize for these rather pedantic stylistic remarks.
As noted above, the authors did a very good job of trying to be extremely detailed yet clear. This is an excellent work.

---

## Round 2 · Author Response

Dear Editor,
We would like to resubmit our manuscript to Scipost. We include all the corrections asked by the two referees. Regarding the questions (2), (3), (4) of the second referee, we have given brief and very partial answers as we feel they are interesting remarks but problems by themselves.
Sincerely yours,
the authors.

---

## Round 2 · List of Changes

Following the first referee's comments. 1) typo corrected. 2) Page 4. We have added that "All loops are fully-packed.". 3) We have updated Figs. 1 and 2 to make them readable in black and white. 4) typo b-2kb-> b+2kb corrected. 5) ponctuation corrected. 6) comma suppressed. 7) double negation suppressed. ---------------------------------------------------------------------------------------- Following the second referee's comments. 1) We have added a separate note on page 2, note [11], "The sites of the kagome lattice are the centers of the edges of the hexagonal lattice." 5) We have rewritten this sentence, page 19. Now it reads "This symmetry implies an equal number...". 6) Singular or plural "dynamics" is left open to proof correction or editor decision. 7) positions of the colons corrected.

---

## Editorial Decision

published